# The effects of assistance dogs on psychosocial health and wellbeing: A systematic literature review

Kerri E. Rodriguez●[1¤]*, Jamie Greer[2], Jane K. Yatcilla[3], Alan M. Beck[1], Marguerite E. O'Haire[1]

**1** Center for the Human-Animal Bond, College of Veterinary Medicine, Purdue University, West Lafayette, Indiana, United States of America, **2** Vassar College, Poughkeepsie, New York, United States of America, **3** Purdue University Libraries, Purdue University, West Lafayette, Indiana, United States of America

¤ Current address: Human-Animal Bond in Colorado, School of Social Work, Colorado State University, Fort Collins, Colorado, United States of America

* kerri.rodriguez@colostate.edu

**Data Availability Statement:** All relevant data are within the paper and its Supporting information files.

## Abstract

Beyond the functional tasks that assistance dogs are trained for, there is growing literature describing their benefits on the psychosocial health and wellbeing of their handlers. However, this research is not only widely disparate but, despite its growth, has not been reviewed since 2012. Our objective was to identify, summarize, and methodologically evaluate studies quantifying the psychosocial effects of assistance dogs for individuals with physical disabilities. Following PRISMA guidelines, a systematic review was conducted across seven electronic databases. Records were independently screened by two authors. Studies were eligible for inclusion if they assessed outcomes from guide, hearing, medical, or mobility service dogs, if they collected original data on handlers' psychosocial functioning, and if the outcome was measured quantitatively with a validated, standardized measure. Studies on psychiatric service dogs, emotional support dogs, and pet dogs were excluded. Of 1,830 records screened, 24 articles were identified (12 publications, 12 theses) containing 27 studies (15 cross-sectional, 12 longitudinal). Studies assessed the effects of mobility (18), hearing (7), guide (4), and medical (2) assistance dog partnerships with an average sample size of N = 83. An analysis of 147 statistical comparisons across the domains of psychological health, quality of life, social health, and vitality found that 68% of comparisons were null, 30% were positive in the hypothesized direction, and 2% were negative. Positive outcomes included significant effects of having an assistance dog on psychological wellbeing, emotional functioning, self-esteem, and vitality. However, it is of note that several methodological weaknesses of the studies make it difficult to draw any definitive conclusions, including inadequate reporting and a failure to account for moderating or confounding variables. Future research will benefit from stronger methodological rigor and reporting to account for heterogeneity in both humans and assistance dogs as well as continued high-quality replication.

**Funding:** The authors received no specific funding for this work.

**Competing interests:** The authors have declared that no competing interests exist.

## Introduction

The roles of dogs to assist in improving human wellbeing continue to expand. Not only are companion dogs prevalent in modern society, but dogs are also often intentionally incorporated into therapeutic processes in the contexts of animal-assisted activities (AAA) and animal-assisted therapy [AAT; 1]. In other contexts, dogs can be specially trained to provide specific benefits to individuals with impairments, disabilities, or chronic conditions as trained assistance animals. Assistance dog placements and roles have grown rapidly in recent decades, especially in the United States, Canada, and Europe [2].

Assistance Dogs International (ADI) defines three types of assistance dogs, of which we use as terminology in this review: guide dogs who assist individuals with visual impairments, hearing dogs who assist individuals with hearing impairments, and service dogs who assist individuals with disabilities other than blindness or deafness [3]. Service dogs can assist individuals with physical disabilities (e.g. performing mobility-related tasks such as pulling a wheelchair or retrieving dropped items), individuals with medical conditions (e.g. alerting or responding to medical crises such hypoglycemia or seizures), and individuals with mental health disorders (e.g. psychiatric service dogs for posttraumatic stress disorder or autism spectrum disorder). Under the Americans with Disabilities Act of 1990, a United States law, an assistance dog must do work or perform tasks for the benefit of an individual with a physical, sensory, psychiatric, intellectual, or other mental disability in order to receive public access rights [4]. While there are no legal requirements specifying that an assistance dog must be certified, registered, or receive any specialized training to receive public access rights, independent organizations such as ADI, the International Association of Assistance Dog Partners (IAADP), and the International Guide Dog Federation (IGDF) define a set of minimum training and behavior standards for public access that help guide the assistance dog industry.

In parallel with an increasing amount of research quantifying the therapeutic benefits of companion dogs and therapy dogs on human health and wellbeing [5, 6], there has been an increased focus on quantifying the physical, psychological, and social effects that assistance dogs may have on their handlers [7–9]. Research has indicated that beyond the physical or tangible benefits that an assistance dog is trained to provide (e.g. route finding, retrieving dropped items, alerting to a seizure), the assistance dog's companionship, emotional and social support, and social facilitation effects in public may be particularly salient to improving the quality of life of individuals with disabilities [7–9]. After receiving an assistance dog, individuals retrospectively report increases to their social, emotional, and psychological health [e.g., 10–12]. Longitudinal studies have found that individuals report improvements to their emotional wellbeing, social functioning, and quality of life just 3 to 6 months after receiving an assistance dog [13–15]. Compared to those on the waitlist, individuals with an assistance dog report better psychosocial functioning and wellbeing [16, 17]. Additionally, research suggests the relationship between an assistance dog and its owner may also serve as a reciprocal attachment and caregiving relationship characterized by secure and strong attachments [18, 19].

To date, there have been several reviews summarizing the literature on the psychosocial effects of assistance dogs on their handlers. One of the first reviews published by Modlin in 2000 [7] summarized nine published quantitative and qualitative studies on the benefits of guide dogs, hearing dogs, and mobility service dogs on their handlers (omitting unpublished theses). Another early review published by Sachs-Ericsson and colleagues in 2002 [8] summarized 14 quantitative studies on both standardized and nonstandardized outcomes following mobility service dog or hearing dog placement (omitting guide dogs). Neither of these early reviews employed a formal methodological assessment of studies, but limitations were listed for each included study. While both reviews found mostly positive findings regarding mobility,

guide, and hearing dogs' effects on their handlers' health and wellbeing, social interactions, and activity participation [7, 8], it was concluded that "the small number of studies and methodological limitations of these studies preclude any clear conclusions" [8].

A more recent systematic review published by Winkle and colleagues in 2012 [9] summarized 12 published quantitative studies on both standardized and nonstandardized outcomes following mobility service dog placement (omitting guide dogs, hearing dogs, and unpublished theses). The scientific rigor of each study was rated according to a 5-level system while the methodological quality of each study was scored on a 7-point scale. While results described positive effects of service dogs in terms of social, psychological, and functional benefits for their handlers, it was concluded that all 12 of the studies had weak study designs with limitations including lack of comparison groups, inadequate description of the service dog intervention, and nonstandardized outcome measures. The authors concluded that although results are promising, "conclusions drawn from the results must be considered with caution" [9].

Because medical service dogs are a relatively new category of assistance dog placements [2], there has been less research on the psychosocial effects of medical alert and response service dogs on their handlers. However, a recent 2018 review summarized five published quantitative studies describing outcomes from seizure alert and seizure response service dogs. The authors found three studies reporting an association between having a seizure alert or response dog and improvements to quality of life and wellbeing, concluding a need for more research.

Research in the field of human-animal interaction (HAI) and assistance dogs is not only rapidly growing but is often disparately published across multidisciplinary journals and outlets. Conducting periodic systematic reviews of this research is crucial to both disseminate knowledge as well as to identify knowledge gaps for future studies [20]. As research on the assistance animal-handler relationship continues to increase, there is a need for an updated, comprehensive collation of the literature encompassing studies on the effects of all varieties of assistance dogs (guide dogs, hearing dogs, and both mobility and medical service dogs) including both published studies and unpublished theses and dissertations. Further, as researchers increasingly incorporate standardized outcome measures into this research, collating and pooling findings will allow researchers to compare outcomes across different populations and interventions while estimating the magnitude of effects across domains.

This research aimed to conduct a systematic assessment of the current state of knowledge regarding the potential benefits of assistance dogs on standardized outcomes of the health and wellbeing of individuals with disabilities. Specifically, this review sought to systematically identify, summarize, and evaluate studies assessing psychosocial outcomes from owning an assistance dog (including service, guide, hearing, and/or medical alert or response dogs) with measures tested for reliability and validity among individuals with physical disabilities. The specific aims were to (1) describe the key characteristics of studies (2) evaluate the methodological rigor of studies (3) summarize outcomes.

## Materials and methods

The systematic literature review was conducted according to The Preferred Reporting Items for Systematic Reviews and Meta-Analyses (PRISMA) guidelines [21]. A study protocol was designed a-priori to define the search strategy, inclusion and exclusion criteria, and items for data extraction.

### Search procedure

As the field of animal-assisted intervention is multidisciplinary, a wide and extensive search was conducted encompassing medical and scientific databases. Further, as publication bias

and the "file-drawer effect" is an often referenced weakness of the HAI literature [22], two dissertation and thesis databases and abstracts of two conferences were searched for unpublished studies.

A health information specialist (JY) constructed and executed comprehensive search strategies in six electronic databases: MEDLINE (PubMed platform), Cumulative Index to Nursing and Allied Health Literature (CINAHL) (EBSCOhost platform), ERIC (EBSCOHost), Web of Science Core Collection (Web of Science), PsycINFO (EBSCOhost), and PsycARTICLES (EBSCOhost). The electronic searches were performed on July 23, 2018, and updated on January 23, 2019. The complete MEDLINE search strategy, which was adapted for the other databases, is shown in S1 Table. Grey literature was addressed by searching ProQuest Dissertations and Theses (ProQuest) and WorldCatDissertations and hand searching the abstracts of the International Society for Anthrozoology and International Association of Human Animal Interactions Organizations conferences.

## Article selection

Studies were eligible for inclusion if they met the following criteria: (1) The study population consisted of current or prospective owners/handlers of an assistance dog (including service, guide, hearing, and/or medical alert or response dogs) with a physical disability or chronic condition in which the assistance dog is trained to do work or perform tasks directly related to the disability or condition [4]; (2) The study collected original data on the effect of the assistance dog on their handler with at least one psychosocial outcome, including those quantifying aspects of mental health, social health, and health-related quality of life; and (3) The psychosocial outcome(s) were collected via a standardized measure tested for validity and reliability. The rationale for excluding studies on emotional service dogs and psychiatric service dogs is that the primary benefits of these dogs are psychological in nature, rather than physical or medical, which complicated comparisons of their psychosocial effects. The rationale for excluding qualitative studies from inclusion was to focus on outcomes using standardized measures to facilitate quantitative comparisons across studies.

## Article screening

All articles were screened by two independent reviewers (authors KR and JG) using Covidence systematic review software (Veritas Health Innovation, Melbourne, Australia). In the case of disagreements, inclusion or exclusion was resolved by discussion and consultation with a third independent reviewer (author MO). After removing duplicate articles in EndNote following a validated protocol [23], articles were screened based on their title and abstract. At this stage, articles were excluded if they were (1) non-English; (2) written for a magazine or other non-peer-reviewed source; (3) book reviews, book chapters, editorials, letters, or opinion papers that did not collect original data; (4) conference abstracts or proceedings; (5) studies assessing companion, therapy, or emotional support animals that were not trained for tasks or work related to a specific disability.

After the initial title and abstract review, articles were screened based on full text. Exclusion criteria were then used to select articles based on the following (in order): (1) irrelevant to study topic; (2) assessed an excluded study population (psychiatric service dogs, therapy dogs, emotional support dogs, or companion dogs); (3) did not report quantitative outcomes from assistance dog placement (literature reviews, instrument development, not original research); (4) reported unrelated outcomes (puppy raising, service dog training, or animal-related outcomes); (5) reported only non-psychosocial outcomes (medical or physical); (6) methodological exclusions (qualitative, case studies, single-subject design); (7) no full text available.

### Data extraction

Articles were extracted for information based on three aims to describe study characteristics, assess methodological rigor, and summarize outcomes. To describe study characteristics, extracted items included participant characteristics (sample size, age, gender, country of origin), assistance dog characteristics (type and provider), and details of the study (design, measurement time points, comparison conditions). To assess methodological rigor, a total of 15 extracted items were sourced from methodological assessment tools including the National Institutes of Health (NIH) Study Quality Assessment Tools [24], the Consolidated Standards of Reporting Trials (CONSORT) checklist [25], the Strengthening the Reporting of Observational Studies in Epidemiology (STROBE) checklists [26], and the Specialist Unit for Review Evidence (SURE) Checklists [27]. Authors JG and KR independently coded 20% of the included articles to establish adequate inter-rater reliability (alpha = 0.822). Author KR then coded 100% of articles. To examine the relationship between methodological rigor score and year of publication as well as sample size, bivariate correlations were performed. To compare methodological rigor by study design, an independent t-test was used to compare mean scores across longitudinal and cross-sectional designs.

To summarize study outcomes, extracted items included statistical comparisons for any psychosocial outcomes from included studies. Because of the broad inclusion criteria, the 27 studies were widely varied in terms of human and dog participants, assessment time points, statistical analyses, and standardized outcomes. Therefore, due to observed heterogeneity, a meta-analysis was not pursued. We also planned to extract or manually calculate effect sizes to create funnel plots to investigate potential publication biases. However, due to large heterogeneity and poor reporting of effect sizes and raw data, a narrative synthesis of findings in comparison to unpublished theses and published articles was pursued instead.

## Results

A total of 1,830 records were screened via title and abstract in which 1,576 records were excluded due to irrelevancy (see Fig 1 for PRISMA diagram). A total of 254 records were screened via full text, of which 230 were excluded. Exclusions included those based on population, outcomes, and methodology. The final sample included 24 articles (12 peer-reviewed publications, 12 unpublished theses/dissertations) containing 27 individual studies. Articles were published from 1994–2018 with publication dates in the 1990s (5), 2000s (9), and 2010s (10) indicating an increasing publication rate on this topic over time.

### Study characteristics

To achieve the first aim of the review–to describe study characteristics–we extracted several features of from each study and article (Table 1).

**Study designs.** Of 27 studies, 15 were cross-sectional and 12 were longitudinal. Studies compared outcomes of individuals with an assistance dog to before they received the dog (six longitudinal studies), to participants on the waitlist to receive an assistance dog (five longitudinal and seven cross-sectional studies), or to participants without an assistance dog (eight cross-sectional studies). Longitudinal assessment time points were varied. Most longitudinal studies (8/12) assessed participants at two time points: at baseline prior to receiving an assistance dog, and an average of 5.8 +/- 3.3 months after participants received an assistance dog (range of 3–12 months follow-up). The remaining four longitudinal studies assessed participants 3–5 times with final follow-up ranging from 9–24 months after receiving an assistance dog.

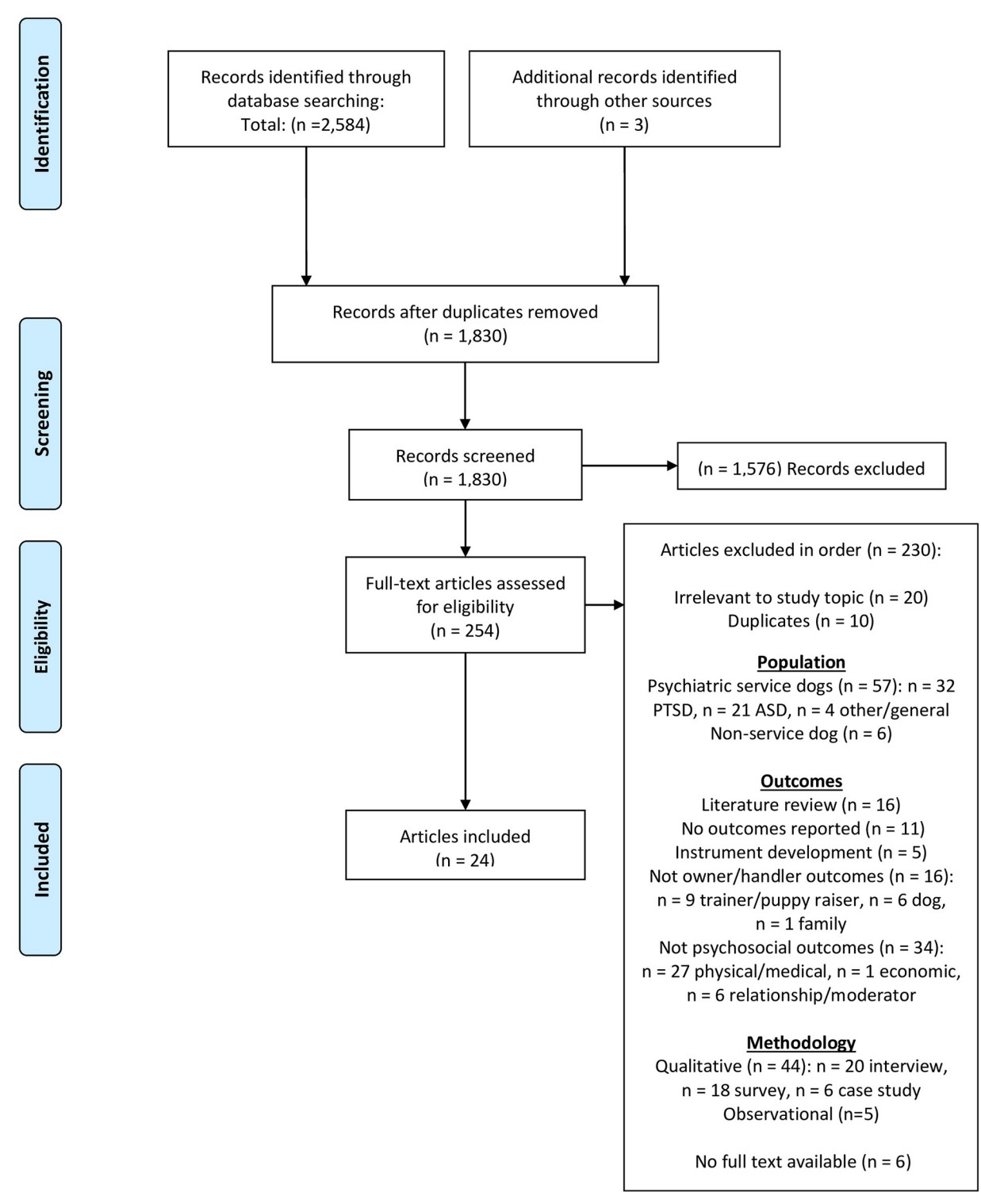

**Fig 1. PRISMA flow diagram.**

**Table 1. Study characteristics of N = 27 studies separated by longitudinal and cross-sectional designs, ordered by publication year.**

| Study, Publication type[a] | Country/ Region | Dog type | Comparison condition | N (treatment/ control) | Participant age (years) | | | % Male participants | Provider organization (s) | Impairments (% total sample) | Assessment time points |
|---|---|---|---|---|---|---|---|---|---|---|---|
| | | | | | M[b] | SD[b] | Rancge[c] | | | | |
| LONGITUDINAL | | | | | | | | | | | |
| Donovan 1994[T] [28] | USA | Mobility | Waitlist; Pre | 52 (26/26) | 35.1 | 10.9 | NR | 50% | CCI | 23% Genetic disability, 45% disability caused by accident, 17% disability caused by illness, 14% CP | T1: 0mo, T2: 4mo |
| Allen & Blascovich 1996[P] [14] | USA | Mobility | Waitlist; Pre | 48 (24/24) | 25 | 1.3 | NR | 50% | NR | 46% SCI, 8% MD, 33% MS, 13% TBI | T1: 0mo, T2: 6mo, T3: 12mo, T4: 18mo, T5: 24mo |
| Gilbey 2003[T], [29] Study #1 | UK | Hearing | Pre | 14 (14/NA) | NR | NR | NR | NR | HDDP | 100% Hearing-impaired | T1: 0mo, T2: 6mo |
| Collins 2004[T] [30] | USA | Mobility | Waitlist; Pre | 20 (11/9) | 42.0 | 11.2 | 18+ | 62% | Paws with a Cause, CCI | 100% Mobility-impaired wheelchair/scooter users | T1: 0mo, T2: 3mo, T3: 9mo |
| Guest et al. 2006[P] [13] | UK | Hearing | Pre | 51 (51/NA) | 51 | NR | 22–87 | 22% | HDDP | 2% Moderate hearing loss, 43% severe hearing loss, 55% profound hearing loss | T1: 0mo, T2: 9.5 +/-6.1mo (end of waiting period), T3: 5 days after T2 (end of 5-day resident training to receive dog), T4: 3.9 +/- 1.4mo after T3, T5: 20.3 +/- 5.4mo after T3 |
| Rabschutz 2006[T] [31] | USA, Canada | Mobility, hearing | Pre | 15 (15/NA) | 46.7 | 14.2 | 29–73 | 33% | NEADS | 33% Deafness, 66% mobility impaired, 20% multiple disabilities | T1: 0mo, T2: 6mo |
| Rintala 2008[P], [32] Study #1 | USA | Mobility | Waitlist; Pre | 33 (18/15) | 47.2 | 12.5 | 21–69 | 24% | THSD, NEADS, PPSD | 64% Quadriplegia, 36% paraplegia | T1: 0mo, T2: 7.09 +/- 0.98mo (treatment), 6.87 +/- 0.50mo (control) |
| Rintala 2008[P], [32] Study #2 | USA | Hearing | Waitlist; Pre | 10 (6/4) | 48.5 | 17.5 | 21–76 | 20% | THSD, NEADS | 90% Severe hearing loss, 10% moderate hearing loss | T1: 0mo, T2: 6.89 +/- 0.61mo (treatment), 6.70 +/- 0.77mo (control) |
| Hubert et al. 2013[P] [33] | Canada | Mobility | Pre | 11 (11/NA) | 32.7 | 12.8 | 18+ | 77% | MIRA Foundation | 38% Paraplegia, 38% quadriplegia, 23% low level spina bifida | T1: 0mo, T2: 7mo |
| Spence 2015[T] [34] | New Zealand | Mobility | Pet Dog; Pre | 17 (7/10) | 49.1 | 13.6 | 21–68 | 35% | MADT | 12% CP; 29% MS, 18% 18% MD, 18% Parkinson's disease, 18% SCI, 6% stroke, 6% other | T1: 0mo, T2: 12mo |
| Vincent et al. 2017[P] [15] | Canada | Mobility | Pre | 17 (17/NA) | 41.9 | 15.3 | 18–64 | 53% | MIRA Foundation | 59% Paraplegia, 24% tetraplegia, 6% leg amputation, 12% CP | T1: 0mo, T2: 3mo, T3: 6mo, T4: 9mo |

*(Continued)*

**Table 1.** (Continued)

| Lundqvist et al. 2018[P] [35] | Sweden | Mobility, hearing, diabetic, seizure | Pre | 55 (55/NA) | 43.8 | 14.0 | 17–68 | 15% | Swedish Association of Service Dogs | 36% Diabetes, 27% neurological, 22% musculoskeletal, 6% deaf/hard of hearing, 4% epilepsy, 5% other | T1: 0mo, T2: 3mo |

| CROSS-SECTIONAL |

| Study, Publication type[a] | Country/ Region | Dog type | Comparison condition | N (treatment/ control) | Participant age (years) | | | % Male participants | Provider organization (s) | Impairments (% total sample) | Time (years) with assistance dog[e] |
|---|---|---|---|---|---|---|---|---|---|---|---|
| | | | | | M[b] | SD | Range[c] | | | | |
| Hacket 1994[T] [36] | USA | Mobility | Waitlist | 40 (24/16) | 37.1 | 10.1 | 21–70 | 43% | Paws with a Cause | 31% SCI, 8% arthritis, 10% CP, 10% MD, 13% MS, 28% Other | M = 1.82, SD = NR, Range = 0.25–4 |
| Rushing 1994[T] [37] | USA | Mobility | Waitlist | 53 (32/21) | 33.4[g] | 7.5[g] | 20–55 | 85% | CCI | 100% Quadriplegia | NR |
| Refson 1999[P] [38] | UK | Guide | No assistance dog | 167 (82/85) | 53 | 76 | 19–94 | 40% | Guide Dogs for the Blind | 100% Visually impaired | M = 9.2, SD = NR, Range = 0.3–45 |
| Gilbey 2003[T] [29] Study #2 | UK | Hearing | Waitlist | 131 (98/33) | 55.4 | 17.3 | NR | 25% | Hearing Dogs for Deaf People | 100% Hearing-impaired | NR |
| Collins et al. 2006[P] [39] | USA | Mobility | No assistance dog | 152 (76/76) | 44.4 | 12.1 | 18+ | 62% | Paws with a Cause, CCI | 41% SCI, 24% non-progressive disability, 34% progressive disability | M = 3.1, SD = NR, Range = 0–13.1 |
| Craft 2007[T] [40] | USA | Mobility | Waitlist | 86 (76/10) | 44.2 | NR | 19–72 | 17% | IAADP, CCI, CST, ADI | 100% Chronic physical disability | NR |
| Milan 2007[T] [41] | USA | Mobility | No assistance dog | 214 (99/ 115) | 44.1 | 12.9 | 18+ | 36% | Paws with a Cause, CCI | 17% Tetraplegia 20% paraplegia, 39% progressive disability, 21% non-progressive disability | M = 3.4, SD = 2.1, Range = 1.2–8.5 |
| Matsunaka & Koda 2008[P] [42] | Japan | Guide | No assistance dog | 80 (30/50) | 34.1 | NR | 15–67 | 55% | Japan Guide Dog Association | 100% Visually impaired | NR |
| Shintani et al. 2010[P] [17] | Japan | Mobility | No assistance dog | 38 (10/28) | 50.0 | 14.0 | 20–67 | 45% | NR | 45% SCI, 26% RA, 11% Stroke, 18% other | M = 1.7, SD = 0.7, Range = 0.7–3.2 |
| Crudden et al. 2017[P] [43] | USA | Guide | No assistance dog | 316 (101/ 215) | 47.7 | 12.3 | 18–65* | NR | NR | 40% Totally blind, 55% legally blind, 6% less severe visual impairment | NR |
| Davis 2017[T] [44] | USA | Mobility | No assistance dog | 140 (91/49) | 41.0 | 14.9 | 18–73 | 40% | AVD, ADW, ADInst, Canine Assistants, CCI, CPL, FSD, HTAD, NEADS, NSD, PPSD | 26% Amputation, 74% neurologically impaired | NR |
| Hall et al. 2017[P] [45] Study #1 | UK | Mobility | Waitlist | 96 (72/24) | NR | NR | 18+ | 20% | Dogs for Good | 30% wheelchair user, 24% MS, 45% other impairments, 5% non-disclosed | NR |

*(Continued)*

**Table 1.** (Continued)

| | | | | | | | | | | | |
|---|---|---|---|---|---|---|---|---|---|---|---|
| Hall et al. 2017[P] [45] Study #2 | UK | Hearing | Waitlist | 141 (111/30) | NR | NR | 18+ | 23% | Hearing Dogs for Deaf People | 100% Hearing-impaired | NR |
| Yarmolkevich 2017[T] [46] | USA | Guide | No assistance dog | 87 (50/37)[h] | NR | NR | 18+ | 61% | Guiding Eyes for the Blind | 52% Totally blind, 12% Near-totally blind, 19% profoundly vision impaired, 9% severely vision impaired, 4% moderately vision impaired, 5% mildly vision impaired | M = NR, SD = NR, Range = 0.5–5+ |
| Rodriguez et al. 2018[P] [16] | USA | Mobility, diabetic, seizure | Waitlist | 154 (97/57) | 26.3 | 17.4 | 4–72 | 53% | Canine Assistants | 26% Seizure disorder, 22% musculoskeletal, 46% neuromuscular, 3% developmental or intellectual, 4% diabetes | M = 4.7*, SD = 3.4*, Range = 0.6–13.7* |

M, Mean; SD, Standard deviation; Pre, Prior to receiving assistance dog; NR = Information not reported;

\* = Information was obtained via email correspondence with a study author.

Provider Organizations: CCI, Canine Companions for Independence; HDDP, Hearing Dogs for Deaf People; NEADS, National Education for Assistance Dog Service; PPSD, Patriot Paws Service Dogs; THSD, Texas Hearing and Service Dogs; MADT, Mobility Assistance Dogs Trust; IAADP, International Association of Assistance Dog Partners; CST, Canine Support Teams; ADI, Assistance Dogs International, Inc.; AVD, America's Veterans Dogs; ADInst, Assistance Dog Institute; CPL, Canine Partners for Life; FSD, Freedom Service Dogs; HTAD, Honor Therapy and Assistance Dogs; NSD, National Service Dogs.

Disabilities/Conditions: SCI, Spinal cord injury; MD, Muscular dystrophy; MS, Multiple sclerosis; TBI, Traumatic brain injury; CP, Cerebral palsy; RA, Rheumatoid Arthritis.

[a] [P], Peer-reviewed publication in an academic journal;

[T], Thesis or dissertation for a Ph.D. or Master's degree.

[b] Values reported to one decimal place unless not reported by authors.

[c] 18+ indicates that authors specified that participants were over 18, but did not provide an upper limit to age range.

[d] Wording used is identical to the original manuscript.

[e] Time since initial assistance dog placement for the treatment/assistance dog group in cross-sectional designs.

[f] Only median age was provided.

[g] Only age values for the treatment group were provided.

[h] Guide dog and guide dog + pet dog groups were collapsed to form the treatment group; Pet dog + no dog groups were collapsed to form the control group.

**Study participants.** Most studies (15/27; 56%) were conducted in the United States, followed by the United Kingdom (6/27; 22%). Other countries where studies took place included Canada (3), Japan (2), New Zealand (1), and Sweden (1). A majority of studies (18/27; 67%) assessed outcomes from mobility service dogs for individuals with physical disabilities. These 18 studies recruited study populations with a range of physical impairments including para- or quadriplegia, musculoskeletal disorders, and neuromuscular disorders. Other studies assessed outcomes from hearing dogs (7/27; 26%), guide dogs (4/27; 15%), and medical alert/response service dogs (2/27; 7%). Human participants in these studies included those with hearing or visual impairments, diabetes, and seizure disorders. Most studies (24/27; 89%) assessed outcomes from a single type of assistance dog (e.g. mobility or guide), thus restricting human participants to a single category of impairments. However, three studies collapsed analyses across several types of assistance dogs and impairments. Most studies (17/27; 63%) recruited from a single assistance dog provider organization, while the remaining studies recruited from a

range of providers (7/27; 26%) or did not report the source of the assistance dogs in the study (3/27; 11%). The most common provider organizations represented were Canine Companions for Independence (CCI; six mobility service dog studies), Paws with a Cause (four mobility service dog studies), and Hearing Dogs for Deaf People (HDDP; four hearing dog studies).

Samples sizes ranged from 10 to 316 participants with an average sample size across all studies of N = 83 +/- 74 participants and a median sample size of N = 53. Seven studies (26%) had sample sizes less than or equal to N = 20, all of which were longitudinal. However, more than half of all studies (16/27; 59%) had sample sizes greater than or equal to N = 50. Cross-sectional studies had the highest sample sizes with an average sample size of N = 126 +/- 73 participants (range of N = 38–316), while longitudinal studies averaged N = 29 +/- 18 participants (range of N = 10–55). Only a single study [16] assessed outcomes from child participants under the age of 18 (an additional study [38] had a minimum inclusion age of 16, but the youngest participant was 19). Average age across all studies was 42 +/- 13 years old. Samples ranged from 15% male to 85% male, with an average of 42% male participants across all studies.

## Study methodologies

To achieve the second aim of the review–to evaluate the methodological rigor of studies–each study was assessed if they met a set of 15 methodological rating items using a scale of yes, no, or N/A (Table 2). Fig 2 displays the total scores across each of the 15 items, separated by introduction, methods, results, and discussion sections (see S2 Table for individual study scores). Overall, studies addressed an average of 62% of methodological consideration items with a range of 23% (3/13) to 100% (15/15; denominators were variable as there were two items not applicable to all study designs). Longitudinal studies addressed an average of 59% of methodological items while cross-sectional studies averaged 65%. However, methodological rigor did

**Table 2. Summary of methodological ratings for N = 27 studies ordered by reporting section (Introduction, Methods, Results, Discussion).**

|  |  | Methodological Rating Item | # of Studies | | |
|---|---|---|---|---|---|
|  |  |  | Yes | No | N/A |
| **Introduction** | Objective | Was an aim, purpose, objective, or research question stated? | 27 | 0 | 0 |
|  | Hypothesis | Was a hypothesis stated? | 17 | 10 | 0 |
| **Methods** | Ethical approval | Was ethical approval for human subjects sought, received, and stated? | 16 | 11 | 0 |
|  | Demographics | Were key demographic characteristics of study participants described including average age and percent of each sex? | 23 | 4 | 0 |
|  | Disabilities | Were details provided regarding participant's disabilities in terms of type/diagnosis, severity, progressiveness, or duration since onset? | 22 | 5 | 0 |
|  | Inclusion/ exclusion | Is there a description of inclusion/exclusion criteria of participants? | 17 | 10 | 0 |
|  | Service dogs | Were the service dogs' source/provider and breeds described? | 5 | 22 | 0 |
|  | Control | Does the design include a control/comparison condition? | 21 | 6 | 0 |
| **Results** | Equal groups | Was there a statistical demonstration that groups or baseline characteristics were equivalent on key demographic variables? [N/A if no control or comparison condition] | 15 | 6 | 6 |
|  | Variability | Does the study provide estimates of variability for most outcomes? | 21 | 6 | 0 |
|  | Statistical values | Were statistical values (e.g. *t*, *F*, *B*) for most outcomes reported? | 12 | 15 | 0 |
|  | Effect sizes | Is an effect size estimate given for most outcomes provided? | 6 | 21 | 0 |
|  | Precise *p* values | Have actual probability values been reported for most outcomes? (e.g. reporting 0.035 rather than reporting <0.05, except when less than 0.001) | 15 | 12 | 0 |
|  | Service dog time | Was time since service dog placement considered for analyses? [N/A for longitudinal studies] | 4 | 11 | 12 |
| **Discussion** | Limitations | Were at least two limitations of the study discussed? | 22 | 5 | 0 |

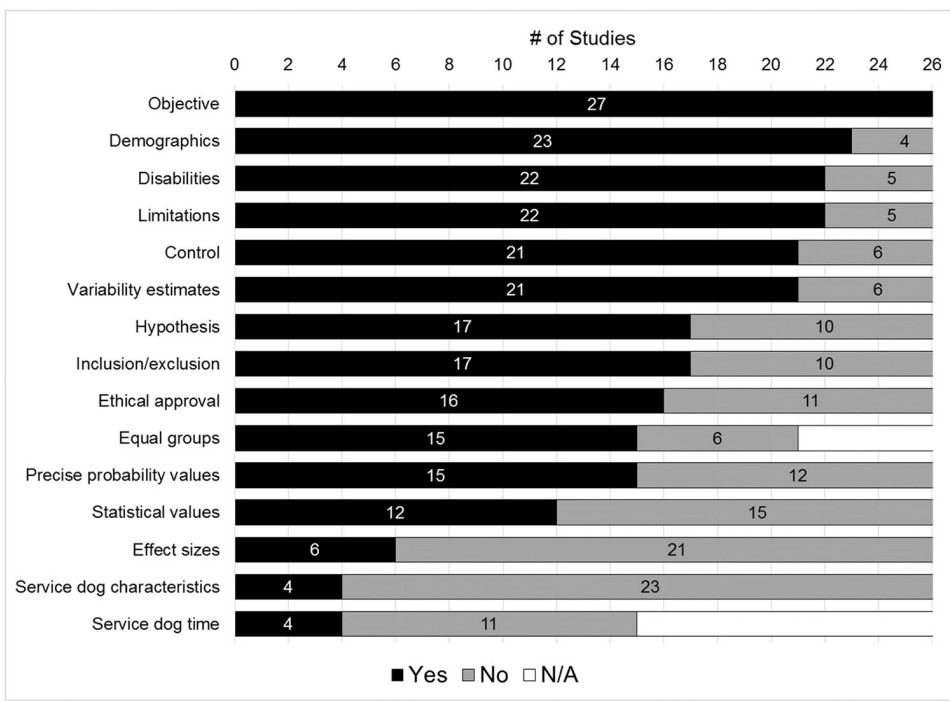

**Fig 2. Visual display of methodological ratings for N = 27 studies ordered by the number of studies addressing each item.**

not significantly differ by study design ($t(25) = -0.940$, $p = 0.356$). Methodological rigor also did not significantly correlate with year of publication ($r = 0.327$, $p = 0.096$) nor total sample size ($r = 0.258$, $p = 0.194$).

In introduction sections, all studies described an objective, but only 17/27 (63%) of studies stated a directional hypothesis. In methods sections, only 16/27 (59%) of studies indicated whether ethical approval for conducting human subjects research was sought and received. Most studies reported adequate detail on participant demographics such as age and sex or gender identity (23/27; 85%) as well as disability characteristics such as primary diagnoses or severity (22/27; 81%). However, inclusion and exclusion criteria were less commonly described (17/27; 63%). Only 5/27 studies (19%) described dogs' breeds and sources. Finally, most studies (21/27;78%) compared outcomes to a control or comparison condition.

In results sections, 15/21 studies with a control or comparison condition (71%) demonstrated that participants in each condition were comparable on demographic variables. This occurred by either matching groups on select criteria or statistically comparing groups' demographic characteristics before performing main analyses. When reporting statistical results, 78% of studies (21/27) provided estimates of variability for outcomes, including confidence intervals, standard deviations, or standard error of the mean. However, only 44% (12/27) of studies reported statistical values (e.g. $t$, $F$, or $B$ values) and only 55% (15/27) of studies reported exact probability values from analyses. Only 6/27 (22%) reported any estimates of effect size in their results. Of 15 cross-sectional studies that surveyed individuals who owned assistance dogs for variable periods of time, 4/15 studies (27%) considered length of time of assistance dog ownership as a potential explanatory or moderating variable in analyses. Finally, in discussion sections, most studies (22/27; 81%) stated at least two limitations of their study.

## Study outcomes

To achieve the third aim of the review–to summarize outcomes–psychosocial outcomes within each study were extracted. Studies made an average of 5.4 statistical comparisons on psychosocial outcomes, ranging from 1–15 comparisons. In total, 147 comparisons were made across the 27 studies that examined the effect of having an assistance dog on a standardized scale or sub-scale on a psychosocial outcome: 58 (39%) psychological outcomes, 43 (29%) social outcomes, 34 (23%) quality of life outcomes, and 12 (8%) energy/vitality outcomes. Of 147 comparisons, 44 (30%) were positive (improved or better functioning in comparison to pre- or control conditions), 100 (68%) were null (no observed difference), and 3 (2%) were negative (decreased or worse functioning in comparison to pre- or control conditions). Of the 44 positive comparisons, 36 (82%) were from published papers and 8 (18%) were from unpublished theses. Of the 100 null comparisons, 43 (43%) were from published papers and 57 (57%) were from unpublished theses.

**Psychological outcomes.** Table 3 summarizes psychological outcomes across studies in terms of general psychological health, emotional health, mental health, and self-evaluation. Of 27 studies, 20 (74%) assessed a psychological outcome with a total of 24 different standardized measures. Of 58 total psychological outcomes, 21 (37%) were positive (improved or better psychological health in comparison to pre- or control conditions), 37 (63%) were null (no difference), and zero (0%) were negative (decreased or worse functioning in comparison to pre- or control conditions).

For general psychological health, 5/11 (45%) outcomes were significant across group or condition. Six studies used standardized measures to assess general health and health symptoms, three of which [17, 28, 35] reported null findings on the general health domain of the RAND 36-Item Short Form Health Survey [SF-36; 47]. However, Lundqvist et al. [35] found increased SF-36 health transition scores after 3-months of having a mobility, hearing, or medical service dog, while Guest [13] found an increase in general health 3-months after receiving a hearing dog using the 30-item General Health Questionnaire [GHQ-30; 48]. Three studies found positive findings on measures of overall psychological wellbeing or psychosocial health, including increased psychological wellbeing 3-months after receiving a mobility, hearing, or medical service dog [35], 6-months after receiving a mobility service dog [14], and better overall psychosocial health in those with a mobility or medical service dog compared to a control group [16]. On the other hand, Spence [34] found no improvement to a composite score of psychological health 12-months after receiving a mobility service dog.

Regarding emotional health, 7/15 (46%) outcomes were significant across group or condition. Yarmolkevich [46] found a significant effect of having a guide dog on positive affect using the Scale of Positive and Negative Experience [SPANE; 49] compared to a control group, while others studies found no effect of having a hearing dog [29] or mobility service dog [39] on affect via the Positive and Negative Affect Scale [PANAS; 50]. Guest [13] used the Profile of Mood States Scale [POMS; 51], finding less overall mood disturbance, less tension, and less confusion 3-months after hearing dog placement. In terms of emotional functioning, two studies found positive results using the SF-36 role emotional domain; Lundqvist et al. [35] found increased functioning 3-months after receiving a mobility, hearing, or medical service dog, while Shintani et al. [17] found better functioning among those with a mobility service dog compared to a control group. On the other hand, Donovan [28] found no change in emotional functioning 4-months after receiving mobility service dog. Using a different measure of emotional functioning, Rodriguez et al. [16] found higher emotional functioning in those with a mobility or medical service dog compared to a control group.

**Table 3. Summary of psychological outcomes across N = 27 studies ordered by sub-category, then by standardized measure.**

| Sub-Category | First author (year) | Standardized Measure | Outcomes (relative to comparison condition) |
|---|---|---|---|
| **General Psychological Health** | Guest (2006) | GHQ-30 | ↑ General health (pre-3mo**, pre-12mo**) |
| | Lundqvist (2018) | SF-36 | — General health (pre-3mo) |
| | Shintani (2010) | SF-36 | — General health (control) |
| | Donovan (1994) | SF-36 | — General health (pre-4mo, control) |
| | Lundqvist (2018) | SF-36 | ↑ Health transition (pre-3mo*) |
| | Gilbey (2003) #1 | SSC | — Health symptoms (pre-6mo) |
| | Gilbey (2003) #2 | SSC | — Health symptoms (control) |
| | Allen (1996) | ABS | ↑ Psychological well-being (pre-6mo***, pre-12mo***, pre-18mo***, pre-24mo***; control***) |
| | Lundqvist (2018) | WHO-5 | ↑ Well-being (pre-3mo*) |
| | Spence (2015) | WHOQOL-BREF | — Psychological health (pre-12mo, control)[a] |
| | Rodriguez (2018) | PedsQL GCS | ↑ Overall psychosocial health (control***) |
| **Emotional Health** | Yarmolkevich (2017) | SPANE | ↑ Positive affect (control) [a] |
| | Gilbey (2003) #1 | PANAS | — Positive affect (pre-6mo) |
| | Gilbey (2003) #2 | PANAS | — Positive affect (control) |
| | Collins (2006) | PANAS | — Positive affect (control) |
| | Gilbey (2003) #2 | PANAS | — Negative affect (control) |
| | Collins 2006 | PANAS | — Negative affect (control) |
| | Guest (2006) | POMS | ↓ Overall mood disturbance (pre-3mo**, pre-12mo**) |
| | Guest (2006) | POMS | ↓ Tension (pre-3mo**, pre-12mo**) |
| | Guest (2006) | POMS | — Aggression (pre-3mo, pre-12mo) |
| | Guest (2006) | POMS | ↓ Confusion (pre-3mo**, pre-12mo**) |
| | Rodriguez (2018) | PROMIS Anger | — Anger (control) |
| | Lundqvist (2018) | SF-36 | ↑ Role emotional (pre-3mo*) |
| | Shintani (2010) | SF-36 | ↑ Role emotional (control**) |
| | Donovan (1994) | SF-36 | — Role emotional (pre-4mo, control) |
| | Rodriguez (2018) | PedsQL GCS | ↑ Emotional functioning (control**) |
| **Mental Health** | Donovan (1994) | SF-36 | — Mental health (pre-4mo, control) |
| | Lundqvist (2018) | SF-36 | — Mental health (pre-3mo*) |
| | Shintani (2010) | SF-36 | — Mental health (control) |
| | Shintani (2010) | SF-36 | ↑ Mental component summary (control**) |
| | Rintala (2008) #1 | SF-12 | — Mental health (pre-7mo, control) |
| | Rintala (2008) #2 | SF-12 | — Mental health (pre-7mo, control) |
| | Milan (2007) | CES-D | — Depression (control) |
| | Collins (2006) | CES-D | — Depressive symptoms (control) |
| | Craft (2007) | CES-D | — Depression (control) |
| | Donovan (1994) | CES-D | — Depression (pre-4mo, control) |
| | Guest (2006) | POMS-SF | ↓ Depression (pre-3mo**, pre-12mo**) |
| | Guest (2006) | GHQ-30 | ↓ Depression (pre-3mo**, pre-12mo**) |
| | Guest (2006) | GHQ-30 | ↓ Anxiety (pre-3mo**, pre- 12mo**) |

*(Continued)*

**Table 3.** (Continued)

| Sub-Category | First author (year) | Standardized Measure | Outcomes (relative to comparison condition) |
|---|---|---|---|
| Self-Evaluation | Allen (1996) | RSES | ↑ Self-esteem (pre-6mo***, pre-12mo***, pre-18mo***, pre-24mo***; control***) |
| | Lundqvist (2018) | RSES | ↑ Self-esteem (pre-3mo*) |
| | Rabschutz (2006) | RSES | ↑ Self-esteem (pre-6mo*) [a] |
| | Yarmolkevich (2017) | RSES | ↑ Self-esteem (control*) [a] |
| | Collins (2006) | RSES | — Self-esteem (control) |
| | Milan (2007) | RSES | — Self-esteem (control) |
| | Hackett (1994) | ISE | — Self-esteem (control) |
| | Donovan (1994) | CSEI | — Self-esteem (pre-4mo, control) |
| | Vincent (2017) | PIADS | — Self-esteem (3mo-6mo, 6mo-12mo, 3mo-12mo) |
| | Vincent (2017) | PIADS | — Adequacy (pre-3mo, pre-6mo, pre-12mo) |
| | Vincent (2017) | PIADS | — Competency (pre-3mo, 3mo-6mo, 6mo-12mo, 3mo-12mo) |
| | Vincent (2017) | RNLI | — Comfort with self (pre-3mo, pre-6mo, pre-12mo) |
| | Refson (1999) | AIS | — Acceptance of disability (control) |
| | Donovan (1994) | ATDP | — Positive attitude towards disability (pre-4mo, control) |
| | Allen (1996) | SCS | ↑ Internal locus of control (pre-6mo***, pre-12mo***, pre-18mo***, pre-24mo***; control***) |
| | Yarmolkevich (2017) | SCCS | — Self-concept clarity (control) [a] |
| | Yarmolkevich (2017) | FS | — Flourishing (control) [a] |
| | Rushing (1994) | TSCS | — Total self-concept (control) |
| | Rushing (1994) | TSCS | — Total positive self-concept (control) |

↑, Increase/Higher;

↓, Decrease/Lower;

***, $p \leq 0.001$;

**, $p \leq 0.01$;

*, $p \leq 0.05$

Standardized Measures: GHQ-30, 30-item General Health Questionnaire; SF-36, RAND 36-Item Short Form Health Survey; SSC, Shortened Symptom Checklist; ABS, Affect Balance Scale; WHO-5, The World Health Organization- Five Well-Being Index; WHOQOL-BREF, World Health Organization Quality of Life Instrument (shortened version); PedsQL GCS, Pediatric Quality of Life Inventory Generic Core Scales; SPANE, Scale of Positive and Negative Experience; PANAS, Positive and Negative Affect Scale; POMS-SF, Profile of Mood States Scale Short Form; PROMIS Anger, Patient-Reported Outcome Measurement Information System Anger Adult Short Form 5A; SF-12, 12-Item Short Form Health Survey; CES-D, Center for Epidemiologic Studies Depression Scale; RSES, Rosenburg Self-Esteem Scale; ISE, Index of Self-Esteem; CSEI, Coopersmith Self-Esteem Inventory; PIADS, Psychosocial Impact of Assistive Devices Scale; RNLI, Reintegration to Normal Living Index; AIS, Felton's Acceptance of Illness Scale; ATDP, Attitudes Towards Disabled Persons Scale; SCS, Spheres of Control Scale; SSCS, Self-Concept Clarity Scale; FS, The Flourishing Scale; TSCS, Tennessee Self-Concept Scale.

[a] Statistical significance was calculated manually via raw data reported in manuscript text.

A total of 13 mental health outcomes were assessed in which 4 (31%) were significant across group or condition. Of 5 studies that used the mental health domain of the SF-36 or the shorter 12-Item Short Form Health Survey (SF-12), only Shintani et al. [17] found an effect of having an assistance dog on mental health. The other four studies reported no changes in participants' mental health 3-months after receiving a mobility, hearing, or medical service dog [35], 4-months after receiving a mobility service dog [28], and 7-months after receiving a hearing or mobility service dog [32]. Six comparisons were made to measure the effect of having an assistance dog on clinical measures of depression or anxiety. However, none of the four studies using the Center for Epidemiologic Studies Depression Scale [CES-D; 52] found significant

differences in self-reported depression among those with a mobility service dog compared to a control group [39–41] or after 4-months with a mobility service dog [28]. However, Guest et al. found significantly lower depression and anxiety using the POMS and GHQ-30, respectively, 6-months after receiving a hearing dog [13].

In the self-evaluation subcategory, 5/19 (26%) outcomes found a significant effect of having an assistance dog on standardized measures of self-esteem, self-concept, and other measures of self-evaluation. Nine studies assessed self-esteem as a primary outcome, with four studies [14, 32, 36, 46] finding a significant effect of having a guide, hearing, mobility, or medical service dog on self-esteem as measured by the Rosenberg Self Esteem Scale [RSES; 53]. However, other studies reported no relationship between having a mobility service dog and self-esteem via the RSES [39, 41] or other standardized measures of self-esteem [15, 28, 36]. Using the Psychosocial Impact of Assistive Devices Scale [PIADS; 54], Vincent et al. [15] found no difference in self-esteem, adequacy, or competency over 12-months following receiving a mobility service dog. Other self-evaluation outcomes assessed with null findings included no differences in self-concept between control groups and those with mobility service dogs [37] or guide dogs [46], no differences in attitude towards a disability 4-months after receiving a mobility service dog [28] or among guide dog users compared to a control group [38], and no differences in flourishing among guide dog users compared to a control group [46]. The only other positive outcome was from Allen et al. [14] which found significantly higher internal locus of control 6-months after receiving a mobility service dog.

**Social outcomes.** Table 4 summarizes the social outcomes across studies within the subcategories of general social functioning, loneliness, and social participation. Of 27 studies, 18 (67%) reported outcomes a standardized measure of social health with a total of 18 different standardized measures. Of 43 total social outcome comparisons, 7 (16%) were positive (improved or better social health in comparison to pre- or control conditions), 36 (84%) were null (no difference) and zero (0%) were negative (decreased or worse social health in comparison to pre- or control conditions).

In terms of general social functioning, 2/10 comparisons made were significant. Three studies using SF-36 failed to find significant effects on the social domain; Lundqvist et al. [35] found no improvement 3-months after receiving a mobility, hearing, or medical service dog, Donovan [28] found no improvement 4-months after receiving a mobility service dog, and Shintani et al. [17] found no difference among mobility service dog users compared to controls. However, on different measures of social functioning Rodriguez et al. found better social functioning in those with a mobility or medical service dog compared to a control group [16] while Guest found improved social functioning 3- and 12-months after receiving a hearing dog [13]. In addition, null findings were reported on standardized measures of family role 3-, 6-, and 12-months after receiving a mobility service dog [15], discrimination and social inclusion 12-months after receiving a mobility service dog [34], and family and social self-concept among mobility dog users compared to a control group [37].

The sub-category of loneliness had 19 comparisons in which only 1/19 (5%) was significant. Of five studies using a version of the UCLA Loneliness Scale [55] only Yarmolkevich [46] found significantly lower self-reported loneliness in those with a guide dog compared to a control group. Four studies found no effect of having a hearing dog [29] or mobility service dog [39, 41] on the UCLA Loneliness Scale. Two studies from the a single thesis [29] made the remaining 14 comparisons on measures of loneliness distress and complementary loneliness, finding no significant changes to loneliness six months after receiving a hearing dog and no significant group differences in loneliness compared to those without a hearing dog.

Regarding social participation, 14 comparisons were made in which 4/14 were significant (29%). Two studies found increased social participation 3-, 6-, and 12-months [15] as well as

**Table 4. Summary of social outcomes across studies ordered by sub-category, then by standardized measure.**

| Sub-Category | First author (year) | Standardized Measure | Outcomes (relative to comparison condition) |
|---|---|---|---|
| **General Social Functioning** | Lundqvist (2018) | SF-36 | — Social functioning (pre-3mo) |
| | Shintani (2010) | SF-36 | — Social functioning (control) |
| | Donovan (1994) | SF-36 | — Social functioning (pre-4mo, control) |
| | Rodriguez (2018) | PedsQL GCS | ↑ Social functioning (control*) |
| | Guest (2006) | GHQ-30 | ↑ Social functioning (pre-3mo**, pre-12mo**) |
| | Vincent (2017) | RNLI | — Family role (pre-3mo, pre-6mo, pre-12mo) |
| | Rushing (1994) | TSCS | — Family self-concept (control) |
| | Rushing (1994) | TSCS | — Social self-concept (control) |
| | Spence (2015) | WHOQOL-DIS | — Discrimination (pre-12mo, control) [a] |
| | Spence (2015) | WHOQOL-DIS | — Social inclusion (pre-12mo, control) [a] |
| **Loneliness** | Gilbey (2003) #2 | UCLA-LS | — Loneliness (control) |
| | Gilbey (2003) #1 | UCLA-LS | — Loneliness (pre-6mo) |
| | Milan (2007) | UCLA-LS | — Loneliness (control) |
| | Yarmolkevich (2017) | UCLA-3 | ↓ Loneliness (control**) [a] |
| | Collins (2006) | UCLA-3 | — Loneliness (control) |
| | Gilbey (2003) #2 | LDS | — Loneliness distress (control) |
| | Gilbey (2003) #1 | LDS | — Loneliness distress (pre-6mo) |
| | Gilbey (2003) #2 | 6-CLS | — Need to keep busy to avoid feeling lonely (control) |
| | Gilbey (2003) #1 | 6-CLS | — Need to keep busy to avoid feeling lonely (pre-6mo) |
| | Gilbey (2003) #2 | 6-CLS | — Need to care for others (control) |
| | Gilbey (2003) #1 | 6-CLS | — Need to care for others (pre-6mo) |
| | Gilbey (2003) #2 | 6-CLS | — Need for tactile affection (control) |
| | Gilbey (2003) #1 | 6-CLS | — Need for tactile affection (pre-6mo) |
| | Gilbey (2003) #2 | 6-CLS | — Need to feel valued and loved (control) |
| | Gilbey (2003) #1 | 6-CLS | — Need to feel valued and loved (pre-6mo) |
| | Gilbey (2003) #2 | 6-CLS | — Belief of being perceived as lonely (control) |
| | Gilbey (2003) #1 | 6-CLS | — Belief of being perceived as lonely (pre-6mo) |
| | Gilbey (2003) #2 | 6-CLS | — Need to share (control) |
| | Gilbey (2003) #1 | 6-CLS | — Need to share (pre-6mo) |
| **Social Participation** | Donovan (1994) | SSBP | — Social participation (pre-4mo, control) |
| | Hubert (2013) | LIFE-H | ↑ Social participation (pre-7mo*) |
| | Vincent (2017) | RNLI | — Participation in recreational activities (pre-3mo, pre-6mo, pre-12mo) |
| | Vincent (2017) | RNLI | ↑ Participation in social activities (pre-3mo*, pre-6mo*, pre-12mo*) |
| | Vincent (2017) | RNLI | — Personal relationships (pre-3mo, pre-6mo, pre-12mo) |
| | Milan (2007) | CHART | — Social integration (control) |
| | Davis (2017) | CHART-SF | — Social integration (control) |
| | Rabschutz (2006) | SCS-R | ↑ Social connectedness (pre-6mo*) [a] |
| | Allen (1996) | CIQ | ↑ Community integration (pre- 6mo***, pre-12mo***, pre-18mo***, pre-24mo***; control***) |
| | Donovan (1994) | SSBP | — Friendship (pre-4mo, control) |
| | Rodriguez (2018) | PROMIS Comp | — Companionship (control) |
| | Spence (2015) | WHOQOL-BREF | — Social relationships (—pre-12mo, control*) [a] |

(*Continued*)

**Table 4.** (Continued)

| Sub-Category | First author (year) | Standardized Measure | Outcomes (relative to comparison condition) |
|---|---|---|---|
| | Matsunaka (2008) | SCLVI | — Conflict stress (control) |
| | Matsunaka (2008) | SCLVI | — Interactions with others (control) |

[↑], Increase/Higher;

[↓], Decrease/Lower;

***, $p \leq 0.001$;

**, $p \leq 0.01$;

*, $p \leq 0.05$

Standardized Measures: SF-36, RAND 36-Item Short Form Health Survey; PedsQL GCS, Pediatric Quality of Life Inventory Generic Core Scales; GHQ-30, 30-item General Health Questionnaire; RNLI, Reintegration to Normal Living Index; TSCS, Tennessee Self-Concept Scale; WHOQOL-DIS, World Health Organization Quality of Life Disability Module; UCLA-LS, UCLA Loneliness Scale; UCLA-3, 3-item version of the UCLA-LS; LDS, Loneliness Distress Scale; CLS, 6-Complementary Loneliness Scales; SSBP, Survey of Social Behavior Patterns; LIFE-H, The Assessment of Life Habits; CHART, Craig Handicap Assessment and Reporting Technique; CHART-SF, Craig Handicap Assessment and Reporting Technique Short Form; SCS-R, Social Connectedness Scale; CIQ, Community Integration Questionnaire; PROMIS Companionship, Patient-Reported Outcome Measurement Information System Companionship Adult Short Form 6A; WHOQOL-BREF, World Health Organization Quality of Life Instrument (shortened version); SCLVI, Stress Checklist for People with Visual Impairments.

[a] Statistical significance was calculated manually via raw data reported in manuscript text.

7-months [33] after receiving a mobility service dog, while Donovan [28] found no change in social participation 4-months receiving a mobility service dog. Other studies found increased social connectedness 3-months after receiving a mobility or hearing dog [31] and increased community integration 6, 12, 18, and 24 months after receiving a mobility service dog [14]. Using the CHART, both Milan [41] and Davis [44] found no group differences in social integration among those with a mobility service dog control groups. Other null findings included no effect of having a guide dog on social conflict stress and interactions with others [42], no improvement in social relationships 12-months after receiving a mobility service dog, and null findings regarding self-reported friendship and companionship with a mobility or medical service dog [16] or 4-months after receiving a mobility service dog [28].

**Quality of life outcomes.** Table 5 displays all quality of life outcomes across studies within the sub-categories of overall quality of life, life satisfaction, and independence. Of 27 studies, 19 (70%) reported outcomes a quality of life measure with a total of 13 different standardized measures used. Of 34 total quality of life outcomes, 9 (26%) were positive (improved or better quality of life in comparison to pre- or control conditions), 22 (65%) were null (no difference) and 3 (9%) were negative (decreased or worse quality of life in comparison to pre- or control conditions).

In the overall quality of life sub-category, 2/8 (25%) comparisons were significant. Lundqvist et al. [35] found higher health-related quality of life 3-months after receiving a mobility, hearing, or medical service dog on one of three measures used [EuroQol Visual Analog Scale; 56]. Hall et al. [45] found higher health-related quality of life among those with a mobility service dog compared to a control group, but not among those with a hearing dog. Other studies found no effect of having a mobility service dog on quality of life including more specific measures such as physical and environmental quality of life [33, 34].

In the next sub-category, six studies assessed life satisfaction outcomes using Satisfaction with Life Scale [SWLS; 57]. However, only 1/6 (17%) found a significant effect, in which Yarmolkevich found higher life satisfaction among those with a guide dog compared to a control group. The other five studies found no effect of having a mobility service dog [32], hearing dog [29, 32], or guide dog [38] on life satisfaction using SWLS.

**Table 5. Summary of quality of life outcomes across studies ordered by sub-category.**

| Sub-Category | First author (year) | Standardized Measure | Outcomes (relative to comparison condition) |
|---|---|---|---|
| **Overall Quality of Life** | Lundqvist (2018) | EQ-VAS | ↑ Health-related quality of life (pre-3mo**) |
| | Lundqvist (2018) | EQ-5D-3L | — Health-related quality of life (pre-3mo) |
| | Lundqvist (2018) | SF-36 | — Health-related quality of life (pre-3mo) |
| | Hall (2017) #1 | QOLS | ↑ Health-related quality of life (control***) |
| | Hall (2017) #2 | QOLS | — Health-related quality of life (control) |
| | Hubert (2013) | QLI | — Quality of life (pre-7mo) |
| | Spence (2015) | WHOQOL-BREF | — Physical quality of life (pre-12mo, control) [a] |
| | Spence (2015) | WHOQOL-BREF | — Environmental quality of life (—pre-12mo, control*) [a] |
| **Life Satisfaction** | Yarmolkevich (2017) | SWLS | ↑ Life satisfaction (control*) [a] |
| | Gilbey (2003) #1 | SWLS | — Life satisfaction (pre-6mo) |
| | Gilbey (2003) #2 | SWLS | — Life satisfaction (control) |
| | Refson (1999) | SWLS | — Life satisfaction (control) |
| | Rintala (2008) #1 | SWLS | — Life satisfaction (pre-7mo, control) |
| | Rintala (2008) #2 | SWLS | — Life satisfaction (pre-7mo, control) |
| **Independence** | Davis (2017) | CHART-SF | ↓ Occupation (control**) |
| | Rintala (2008) #2 | CHART | ↓ Occupation (pre-7mo*,—control) [b] |
| | Rintala (2008) #1 | CHART | — Occupation (pre-7mo, control) |
| | Milan (2007) | CHART | — Occupation (control) |
| | Davis (2017) | CHART-SF | ↓ Economic self-sufficiency (control*) |
| | Collins (2004) | CHART | — Economic self-sufficiency (pre-3mo, pre-9mo; control) |
| | Milan (2007) | CHART | — Economic self-sufficiency (control) |
| | Davis (2017) | CHART-SF | — Mobility (control) |
| | Milan (2007) | CHART | ↑ Mobility (control) |
| | Rintala (2008) #1 | CHART | — Mobility (pre-7mo, control) |
| | Rintala (2008) #2 | CHART | — Mobility (—pre-7mo, control*) |
| | Matsunaka (2008) | SCLVI | ↓ Mobility stress (control*) |
| | Crudden (2017) | TSS | ↓ Walking stress (control*) |
| | Crudden (2017) | TSS | — Public transportation stress (control) |
| | Hubert (2013) | RNLI | ↑ Ability to return to normal life (pre-7mo*) |
| | Vincent (2017) | RNLI | — Self-care (pre-3mo, pre-6mo, pre-12mo) |
| | Vincent (2017) | RNLI | ↑ Daily work activities (pre-3mo†,—pre-6mo, pre-12mo†) |
| | Vincent (2017) | RNLI | — Ability to deal with life events (pre-3mo, pre-6mo, pre-12mo) |
| | Rodriguez (2018) | PedsQL GCS | ↑ Work/school functioning (control***) |
| | Craft (2007) | A-IIRS | — Perceived intrusiveness of disability (control) |

[↑], Increase/Higher;

[↓], Decrease/Lower;

***, $p \leq 0.001$;

**, $p \leq 0.01$;

*, $p \leq 0.05$;

[†], $p > 0.017$ but $< 0.10$.

Standardized Measures: EQ-VAS, EuroQol visual analogue scale; EQ-5D-3L, EuroQol EQ-5D-3L; SWLS, Satisfaction with Life Scale; SF-36, RAND 36-Item Short Form Health Survey; QOLS, Flanagan Quality of Life Scale; QLI, Quality of Life Index; WHOQOL-BREF, World Health Organization Quality of Life Instrument (shortened version); RNLI, Reintegration to Normal Living Index; SWLS, Satisfaction with Life Scale; CHART, Craig Handicap Assessment and Reporting Technique; CHART-SF, Craig Handicap Assessment and Reporting Technique Short Form; SCLVI, Stress Checklist for People with Visual Impairments; TSS, Transportation Stress Survey; RNLI, Reintegration to Normal Living Index; PedsQL GCS, Pediatric Quality of Life Inventory Generic Core Scales; A-IIRS, Adapted Illness Intrusiveness Ratings Scale.

[a] Statistical significance was calculated manually via raw data reported in manuscript text.

[b] Both the experimental and control groups had lower (worse) occupation scores at follow-up than at baseline.

In the sub-category of independence, a total of 20 comparisons were made in which 9 (45%) were significant, but 3 (15%) were in the negative direction. The most commonly used measure was the Craig Handicap Assessment and Reporting Technique [CHART; 58] which assesses how people with disabilities function as active members of their communities. Using the occupation domain of the CHART, Rintala et al. [32] found no difference in occupational functioning 7-months after receiving a mobility service dog and Milan [41] found no group difference in those with and without a mobility service dog. However, 2 studies found *worse* occupational functioning in terms of employment, schooling, or homemaking. Rintala et al. [32] found that participants reported worse occupational functioning 7-months after receiving a hearing dog while Davis [44] found that individuals with a mobility service dog reported worse occupational functioning compared to a control group.

In the economic domain of the CHART, which assesses socio-economic independence, Davis [44] again found that those with a mobility service dog reported worse economic functioning than controls while two mobility dog studies reported null findings [30, 41]. In the mobility domain, only Milan [41] found a significant effect of having a mobility service dog on the CHART mobility domain (which includes hours per day out of bed and days per week out of the house) while Davis [44] and Rintala et al. [32] reported no relationship between the mobility domain and having a service dog or hearing dog. Using other standardized measures of independence, Matsunaka & Koda [42] found that those with guide dogs reported and lower stress while being mobile. Similarly, Crudden et al. [43] found that individuals who had guide dogs reported less stress while walking, but not while using public transportation. Using the Reintegration to Normal Living Index [RNLI; 59], Hubert found improvements in the ability to return to 'normal life' after 7-months with a mobility service dog while Vincent et al. [15] found improvements to daily work activities 3- and 12-months after receiving mobility service dog (but not in self-care or dealing with life events). Finally, Rodriguez et al. [16] found that those with a mobility or medical service dog reported significantly higher work/school functioning than a control group.

**Vitality outcomes.** Table 6 summarizes vitality outcomes across studies within the sub-categories of general energy/vitality and sleep. Of 27 studies, 7 (26%) reported outcomes from at least one standardized measure of vitality with a total of five different standardized measures. Of 12 total vitality comparisons, 6 (50%) were positive (improved or better vitality in comparison to pre- or control conditions), 6 (50%) were null (no difference) and zero (0%) were negative (decreased or worse vitality in comparison to pre- or control conditions).

In terms of general vitality and energy, four studies used the SF-36 to measure the effect of having an assistance dog on the vitality domain. Only Vincent et al. [15] found a significant increase in pep, energy, and feeling less worn out 3- and 6-months after receiving a mobility service dog while three studies found no relationship between the vitality domain and having a mobility service dog [17, 28] or a mobility, hearing, or medical service dog [35]. Using the Profile of Mood States Scale [POMS; 51], Guest found increased self-reported vigor 3- and 12-months after receiving a hearing dog and less fatigue 3-months after receiving a hearing dog. Using another measure of energy and fatigue, Craft [40] found no difference in those with or without a mobility service dog. Regarding sleep, Guest found better self-reported sleep quality 3- and 12-months after receiving a hearing dog while Rodriguez et al. [16] found no difference in sleep disturbance between individuals with mobility or medical service dog and a control group.

## Discussion

This systematic review summarized the current state of knowledge regarding the effects of owning an assistance dog (including service, guide, hearing, and/or medical alert or response

**Table 6. Summary of vitality outcomes across studies ordered by sub-category, then by standardized measure.**

| Sub-Category | First author (year) | Standardized Measure | Outcomes (relative to comparison condition) |
|---|---|---|---|
| **Energy/Vitality** | Donovan (1994) | SF-36 | — Vitality (—pre-4mo, control*) |
| | Lundqvist (2018) | SF-36 | — Vitality (pre-3mo) |
| | Shintani (2010) | SF-36 | — Vitality (control) |
| | Vincent (2017) | SF-36 | ↑ Pep (pre-3mo*,—pre-6mo,—pre-12mo) |
| | Vincent (2017) | SF-36 | ↑ Energy (pre-3mo*,—pre-6mo,—pre-12mo) |
| | Vincent (2017) | SF-36 | ↓ Worn out (pre-3mo†, pre-6mo*, pre-12mo†) |
| | Vincent (2017) | SF-36 | — Tiredness (pre-3mo, pre-6mo, pre-12mo) |
| | Guest (2006) | POMS | ↑ Vigor (pre-3mo**, pre-12mo**) |
| | Guest (2006) | POMS | ↓ Fatigue (pre-3mo**,—pre-12mo) |
| | Craft (2007) | EFS | — Energy/Fatigue (control) |
| **Sleep** | Guest (2006) | GHQ-30 | ↑ Sleep (pre-3mo**, pre-12mo**) |
| | Rodriguez (2018) | PROMIS SD | — Sleep disturbance (control) |

↑, Increase/Higher;

↓, Decrease/Lower;

***, $p \le 0.001$;

**, $p \le 0.01$;

*, $p \le 0.05$;

†, $p > 0.017$ but $< 0.10$.

Standardized Measures: SF-36, RAND 36-Item Short Form Health Survey; POMS, Profile of Mood States Scale; EFS, Energy/Fatigue Scale; GHQ-30, 30-item General Health Questionnaire; PROMIS SD, Patient-Reported Outcome Measurement Information System Sleep Disturbance Adult Short Form 6A.

dogs) on standardized outcomes of psychosocial health and wellbeing of individuals with disabilities. Our search procedure identified 24 articles containing 27 studies assessing psychosocial outcomes from a wide variety of human and assistance dog populations. These studies were reviewed to complete three specific aims: to describe the key characteristics of studies, to evaluate the methodological rigor of studies, and to summarize outcomes. The discussion section aims to review the findings from each aim and to provide targeted suggestions for future research.

## Study characteristics

Our first aim was to describe study characteristics of the literature. We found that most studies were conducted in either the United States or the United Kingdom, but there was international representation of the research in Canada, Sweden, New Zealand, and Japan. Most articles were published in the 2010s, indicating an increasing publication interest in this topic over time. In fact, nine new articles were identified (three theses, six publications) that had been published since the last review on this topic in 2012 [9]. Increased research on this topic is likely in parallel with the increased roles and demands for different types of assistance dogs worldwide [2] as well as increased interest in the benefits of animal interaction for human health and wellbeing [60]. The most commonly studied type of assistance dog was mobility service dogs, followed by hearing dogs. Guide dogs were only assessed in four studies (all of which were cross-sectional, and one of which was an unpublished thesis [46]). The lack of guide dog-specific research is especially surprising given that guide dogs not only have the longest history of any type of assistance dog [61] but are also the most commonly placed assistance dog placed by professional facilities worldwide [2]. Future longitudinal research in this population is necessary to understand the complex psychosocial and physical roles that guide dogs play in the

lives of their handlers. Medical service dogs for diabetes and seizure alert/response were rarely studied [16, 35], and were assessed in conjunction with mobility service dogs rather than on their own. However, these are relatively new categories of assistance dogs [2], many of which may also be self-trained [62], and it appears that emerging research on this population has centered on medical benefits [63] rather than psychosocial. Future research should focus on assessing outcomes from these medical alert and response assistance dogs and how their roles may be similar or different than mobility, guide, or hearing dogs.

Study designs included both cross-sectional and longitudinal studies, with only one randomized longitudinal study identified [14]. However, it should be noted that this study by Allen & Blascovich has received considerable critique due to incredibly large effect sizes, unrealistic retention and response rates, and severe methodological omissions including a lack of reporting on recruitment, funding, or where assistance dogs were sourced and trained [despite repeated requests for clarification; 64, 65]. The remaining studies were quasi-experimental in that they did not use randomized assignment to treatment or control groups. Therefore, the current literature is limited to correlational, rather than causal conclusions regarding the benefits of assistance dogs on the psychosocial health of their owners. Overall, sample sizes were higher than what is usually observed in targeted animal-assisted intervention studies with dogs (e.g. [66, 67]) but smaller than that of pet dog research [68]. Interestingly, only one included study [16] assessed outcomes from participants under the age of 18. Although outcomes from assistance dog placement for children and adolescents have been quantified with qualitative [e.g., 69–71] and observational [e.g. 72] study designs, effects on standardized measures of psychosocial wellbeing including social functioning have not been explored. Therefore, future studies are warranted that specifically assess health and wellbeing using validated parent-proxy or self-report measures to fully understand the potential effects that assistance dogs can have on children and adolescents with disabilities.

## Methodological rigor

Our second aim was to evaluate the methodological rigor of studies. We found that similar to the range of study characteristics observed, there was considerable variation in the methodological rigor of included studies. The most notable weaknesses included a lack of adequate reporting in the methodological sections, which not only limits interpretation of findings but prevents reproducibility. First, only 59% of studies stated whether ethical approval for human subjects was sought and received. Future research should specify not only ethical protocols for human subjects research, but also for animal subjects, which is often underutilized and/or underreported in AAI research [73]. Second, only 63% of studies described inclusion and/or exclusion criteria of recruited participants, and some studies did not report all demographic or disability characteristics of participants. Future studies should provide detailed researcher-specified criteria for participation as well as organizational-specified criteria for placing/receiving an assistance dog, if applicable. For example, organizations that place assistance dogs may have housing, familial, physical, or even financial requirements for potential recipients that should be subsequently reported in the manuscript to fully define the population. It is unreasonable to assume that the changes to an individual's life following receipt of an assistance dog is identical for all ages, gender identities, backgrounds, and disabilities. Therefore, detailed descriptions of study populations is critical for helping the field understand for whom assistance dogs are beneficial regarding social, emotional, or psychological health and under what contexts or conditions [74].

Finally, one of the most notable examples of poor methodological reporting across studies was the omission of information regarding assistance dogs' sources (e.g. purpose-bred from a

provider, self-trained) and breeds (e.g., Labrador Retriever, Golden Retriever, Mixes). As the assistance dog itself is the key component of the intervention, details regarding the dog's breeding, rearing, selection, and training, as well as the assistance dog-handler matching process are critical to disentangling potential mechanisms [75]. In addition, reporting detailed information on assistance dogs allows for the consideration of the dogs as individual agents in the therapeutic process rather than as uniform tools [1, 74].

In addition to poor methodological reporting, many studies were restrained by statistical weaknesses. Many studies did not confirm that participants across groups were statistically equivalent on key demographic variables such as age and sex/gender before conducting statistical analyses. This poses a severe threat to the validity of findings as group differences in outcomes could be caused by underlying differences in certain demographics or characteristics and cannot be confidently attributed to the presence of the assistance dog. Secondly, many studies did not report sufficient detail in results in terms of estimates of variability and effect size. Thorough reporting in terms of the magnitude and variability of effects observed will allow researchers to make informed comparisons across populations and interventions and conduct critically needed meta-analyses in the field.

## Study outcomes

The third aim of the review was to summarize psychosocial outcomes of studies. We found that studies reported mostly psychological outcomes (74%), followed by social outcomes (67%), quality of life outcomes (70%), and vitality (26%) outcomes. Overall, most (68%) of comparisons made across studies were null in which no statistical difference was found in the outcome compared to before getting an assistance dog or compared to a control group. Importantly, only a few comparisons were made in the negative direction (2%) indicating that there is limited reason to believe that acquiring an assistance dog is associated with *worse* functioning. A total of 30% of comparisons made were positive in which having an assistance dog was associated with improved psychosocial functioning among individuals with disabilities. In fact, positive findings were identified in all domains and sub-domains of psychosocial health and wellbeing. Promising areas include psychological wellbeing, emotional wellbeing, and social participation in which several positive outcomes were identified. However, almost all positive findings were accompanied by a null finding using the same or similar standardized measure in a different study. The below discussion considers various potential explanations for the inconsistencies in findings across studies.

**Variability in assessment times.** One of the main considerations in understanding the potential variability across findings is the aspect of time since assistance dog placement. In longitudinal studies, the first follow-up time point varied from 3- to 12-months after receiving an assistance dog. Within cross-sectional studies, number of years since first partnering with an assistance dog ranged from 6-months to 45 years with means ranging from 2–9 years. This variation in assessment times makes it difficult to draw definitive conclusions on conflicting findings. Further, the number of years spent with the assistance dog at the time of surveying was unknown for half of the cross-sectional studies [29, 37, 40, 42–45]. Therefore, in the cases where positive outcomes were reported in these studies, it is unknown what amount of time with an assistance dog the finding was associated with (and therefore difficult to compare to findings from other studies).

**Variability in interventions.** Another potential explanation for inconsistent findings across studies lies in the inherent variability of the assistance dog intervention itself. Assistance dog categories (guide, hearing, mobility, and medical) were collapsed for the purposes of this review, but undoubtedly contribute to the lives of individuals with disabilities in diverse ways.

However, even within a single category, there are differences in assistance dog breeds, temperaments, and training that may significantly contribute to observed variance across studies. Second, there is inherent variation in both the quality and quantity of interactions from one assistance dog-owner pair to the next. In addition to the different human and dog phenotypes that contribute to this heterogeneity, there are likely differences in the strength of the human-animal bond and attachment relationships formed between assistance dogs and handlers [19, 76]. Moderator analyses will be useful in determining the potential explanatory effects that handler-service dog relationships have on psychosocial outcomes.

**Variability in standardized measures.**　Another potential reason for the inconsistencies in findings from studies assessing the same construct is disparities across standardized measures. Measures of the same outcome not only can have different wording and items, but also can measure functioning over different time periods or contexts. In one example, four studies included in this review failed to find significant results in comparisons of depression using the CES-D [28, 39–41]. However, positive findings were found in depression using the POMS by a different study [13]. The CES-D asks participants to rate how often they had experienced 20 depressive symptoms in the prior week using statements such as "I thought my life had been a failure," while the POMS asks participants to rate from not at all to extremely how they feel *right now* using single words such as "sad" and "unhappy." It is also possible that some standardized measures do not capture the intended effects from having an assistance dog. One author argued that an "important methodological issue is the absence of appropriate measures" in measuring the effect of an assistance dog on recipients' lives [32]. Future research is necessary to determine if in fact some measures are inappropriate to measure change following an assistance dog, which may be addressed using interviewing and focus group techniques among assistance dog handlers. The replicated measures identified in this review can serve as a basis for future researchers to collate the existing literature when making assessment choices.

**Variability in study rigor.**　A final potential reason for outcome discrepancies is variation in methodological rigor across studies. In particular, not only did studies vary largely in terms of sample size, but they also varied in the manner in which statistical analyses were conducted. As mentioned above, a surprisingly high number of studies did not ensure that assistance dog and control groups were statistically equal across demographic and disability characteristics prior to outcome analyses. In these studies, positive findings (i.e., better social functioning in those with an assistance dog compared to a control group) may be partially attributed to an unmeasured variable driving the group difference [77]. In addition, many studies did not account for confounding variables such as having a pet dog, the progressiveness or type of disability, or relationship status.

**Other considerations.**　An important finding from this review was that most positive findings were reported in published studies, while unpublished theses were more likely to report null findings. This pattern suggests a potential publication bias present in which disproportionately more positive findings are in the published studies than the unpublished theses [78]. Importantly, unpublished theses had a similar average sample size as published studies, with similar power to detect effects compared to published studies. Thus, this pattern may be better explained by the "file drawer effect" in which there is a bias towards publishing positive findings over null findings [79]. Although this tendency occurs in many fields, the file-drawer bias may especially be prevalent in human-animal interaction research due to the preconceived notion that animals are beneficial for humans [80]. In fact, positive, null, and negative findings are equally instrumental in understanding the complexities of the role that assistance dogs play in the lives of individuals with physical disabilities. As Serpell and colleagues point out, individuals that don't benefit from animal-assisted interventions may be just as informative from a scientific perspective as the ones that do, and "the entire field potentially suffers when these

sorts of contrary or ambiguous findings get buried or ignored" [74]. Therefore, future efforts should be made to publish null findings in peer-reviewed journals and to encourage scientific transparency [80].

As a final consideration, it is possible that assistance dogs may not confer significant psychosocial benefits as quantified by some of the standardized measures used. First, there may be ceiling effects present whereby individuals are functioning at initially healthy levels of the measured construct (e.g., depression, self-esteem) prior to receiving an assistance dog and thus may not significantly improve on these measures. This effect may be compounded by the possibility that those who apply for an assistance dog may inherently have certain positive characteristics (e.g., stable housing, stable finances, has a familial support system) that contribute to overall psychosocial health. Further, in contrast to a psychiatric service dog or an emotional support dog, the assistance dogs in this review are not explicitly trained for mental health-related support and their effects on the psychosocial health of their handlers may be variable rather than population-wide. For example, the benefits of an assistance dog for a socially isolated individual who experiences periodic anxiety and depression may be significantly different than an individual without these characteristics. An important question for the field moving forward will be to determine for whom an assistance dog may confer the most significant psychosocial health benefits for, and under what contexts or conditions.

## Conclusions

This systematic review identified 24 articles containing 27 studies that assessed a psychosocial outcome of having an assistance dog (guide dog, hearing dog, mobility service dog, or medical service dog). Included studies assessed psychosocial outcomes via standardized measures from assistance dogs that were trained for functional tasks related to a physical disability or medical condition (omitting psychiatric service dogs or emotional support dogs). Despite the purpose of these assistance dogs specifically for physical tasks, positive outcomes were noted in psychological, social, quality of life, and vitality domains. However, results suggested that for most outcomes, having an assistance dog had no effect on psychosocial health and wellbeing. Methodological weaknesses including poor reporting of assistance dog interventions and statistical limitations prevent any clear conclusions made regarding the psychosocial effects of assistance dogs on individuals with disabilities. Inconsistencies in findings were discussed in terms of wide variability in assessment times, interventions, measures, and rigor, and recommendations were made to contribute to the knowledge of this growing application of the human-animal bond. Continued efforts are required to improve methodological rigor, conduct replicable research, and account for heterogeneity in both humans and animals to advance the state of knowledge in this field.

## Supporting information

**S1 Checklist. PRISMA 2009 checklist.**
(DOC)

**S1 Table. MEDLINE search terms and search strategy.** The search strategy was adapted to the other databases, including mapping terms to each database's thesaurus or prescribed vocabulary, as appropriate.
(DOCX)

**S2 Table. Summary of methodological rating scores by each of the N = 27 individual studies.** Studies are organized by design (longitudinal or cross-sectional).
(DOCX)

## Author Contributions

**Conceptualization:** Kerri E. Rodriguez, Alan M. Beck, Marguerite E. O'Haire.

Data curation: Kerri E. Rodriguez, Jamie Greer, Jane K. Yatcilla, Marguerite E. O'Haire.

**Formal analysis:** Kerri E. Rodriguez, Jamie Greer.

**Investigation:** Kerri E. Rodriguez, Jamie Greer, Jane K. Yatcilla.

**Methodology:** Kerri E. Rodriguez, Jamie Greer, Jane K. Yatcilla.

**Project administration:** Kerri E. Rodriguez, Marguerite E. O'Haire.

**Supervision:** Kerri E. Rodriguez, Alan M. Beck, Marguerite E. O'Haire.

**Writing – original draft:** Kerri E. Rodriguez.

**Writing – review & editing:** Kerri E. Rodriguez, Jamie Greer, Jane K. Yatcilla, Alan M. Beck, Marguerite E. O'Haire.

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
