## [Decision Letter · Decision Letter 0]

15 Oct 2020

PONE-D-20-22650

The effects of assistance dogs on psychosocial health and wellbeing: A systematic literature review

PLOS ONE

Dear Dr. Rodriguez,

Thank you for submitting your manuscript to PLOS ONE. After careful consideration, we feel that it has merit but does not fully meet PLOS ONE’s publication criteria as it currently stands. Therefore, we invite you to submit a revised version of the manuscript that addresses the points raised during the review process.

We look forward to receiving your revised manuscript.

Kind regards,

Geilson Lima Santana, M.D., Ph.D.

Academic Editor

PLOS ONE

Journal Requirements:

Reviewers' comments:

Reviewer's Responses to Questions

**Comments to the Author**

1. Is the manuscript technically sound, and do the data support the conclusions?

Reviewer #1: Yes

Reviewer #2: Yes

2. Has the statistical analysis been performed appropriately and rigorously? 

Reviewer #1: Yes

Reviewer #2: I Don't Know

3. Have the authors made all data underlying the findings in their manuscript fully available?

Reviewer #1: Yes

Reviewer #2: Yes

4. Is the manuscript presented in an intelligible fashion and written in standard English?

Reviewer #1: Yes

Reviewer #2: Yes

5. Review Comments to the Author

Reviewer #1: Title of article: The effects of assistance dogs on psychosocial health and wellbeing: A systematic literature review

Reviewer Comments

1-Regarding Research Question 1: researchers should be consistent in using terms. Problem that they used different terms: Sometime describe, identify, another time, summarize for question 1.

Authors should determine the appropriate terms for three questions, keeping using the same terms in the whole paper.

2-In Discussion Section, authors are clear in mentioning the third aim. I don’t read or see: first aim and second aim. Authors should be add it

3 -Based on my experiences for teaching the systematic review, systematic researchers should provide their manuscript by official or scientific paragraph terms containing terminologies of subjects of systematic reviews that they use it in search database. Why did not researchers mention it at their article? All readers prefer to read and see that paragraph / terms used in journals. Example

(Assistance or support or helping Dogs and ….etc.)

4-(e.g. 0.035 rather than <0.05) why did researchers choose 0.035 as criterion? In Table 2. Summary of methodological ratings for N=27 studies … etc.

The subject of paper is viable, novelty and interested. Systematic review is hard working as I see that in this paper. So, in my opinion paper is acceptable with modification.

Best regard

Naser

Reviewer #2: Have the authors made all data underlying the findings in their manuscript fully available?

N/A

This article is considering and dealing with an excellent subject which is not only sensitive but also helpful. The authors have thought and looked into the psychological benefits of guide dogs and the importance of a holistic approach on such issues. I am hoping by publishing this article, we would expect to see further research work from the same authors and others.

6. PLOS authors have the option to publish the peer review history of their article (what does this mean?). If published, this will include your full peer review and any attached files.

Reviewer #1: **Yes: **Dr. Naser Abdul Hafeeth Abdulhafath. University of Taiz

Reviewer #2: **Yes: **Lily Abedipour

---

## [Author Response · Author response to Decision Letter 0]

20 Oct 2020

Thank you for the opportunity to submit a revised version of this systematic review manuscript. We have uploaded a response to reviewers document, but have also copied and pasted the editor and reviewer's feedback and our responses below. 

Editor Comments

We have fixed formatting on all tables, figures, and table/figure captions. We have also updated the title page formatting. 

We have provided the supporting information captions at the bottom of the manuscript as requested, and updated all in-text citations for the S1 and S2 tables. 

Reviewer 1

1-Regarding Research Question 1: researchers should be consistent in using terms. Problem that they used different terms: Sometime describe, identify, another time, summarize for question 1.

Authors should determine the appropriate terms for three questions, keeping using the same terms in the whole paper.

We have edited the text throughout the paper such that each of the three aims are consistent with the following wording: “The specific aims were to (1) describe the key characteristics of studies (2) evaluate the methodological rigor of studies (3) summarize outcomes.”

2-In Discussion Section, authors are clear in mentioning the third aim. I don’t read or see: first aim and second aim. Authors should be add it

Thank you for this suggestion. In addition to the headers stating each aim, we also start each section with the appropriate language e.g. “Our second aim was to evaluate the methodological rigor of studies. We found that…”

3 -Based on my experiences for teaching the systematic review, systematic researchers should provide their manuscript by official or scientific paragraph terms containing terminologies of subjects of systematic reviews that they use it in search database. Why did not researchers mention it at their article? All readers prefer to read and see that paragraph / terms used in journals. Example (Assistance or support or helping Dogs and ….etc.)

We provided the full MEDLINE search strategy in S1 Table, rather than in the text, so that the search strategy could be fully replicated by other researchers. We describe this in lines 167-169: “The complete MEDLINE search strategy, which was adapted for the other databases, is shown in S1 Table.” The table describes the full search terms used, as follows: 

( “Service animal”[Title/Abstract] OR “service animals”[Title/Abstract] OR “Service dog”[Title/Abstract] OR “Service dogs”[Title/Abstract] OR “Assistance animal”[Title/Abstract] OR “Assistance animals”[Title/Abstract] OR “Assistance dog”[Title/Abstract] OR “Assistance dogs”[Title/Abstract] OR “Guide dog”[Title/Abstract] OR “Guide dogs”[Title/Abstract] OR “Dog guide”[Title/Abstract] OR “Dog guides”[Title/Abstract] OR “Mobility dog”[Title/Abstract] OR “Mobility dogs”[Title/Abstract] OR “Seizure dog”[Title/Abstract] OR “Seizure dogs”[Title/Abstract] OR “Seizure alert dog”[Title/Abstract] OR “Seizure alert dogs”[Title/Abstract] OR “Seizure response dog”[Title/Abstract] OR “Seizure response dogs”[Title/Abstract] OR “Epilepsy alert dog”[Title/Abstract] OR “Epilepsy alert dogs”[Title/Abstract] OR “Diabetes alert dog”[Title/Abstract] OR “Diabetes alert dogs”[Title/Abstract] OR “Diabetic alert dog”[Title/Abstract] OR “Diabetic alert dogs”[Title/Abstract] OR “Diabetic response dog”[Title/Abstract] OR “Diabetic response dogs”[Title/Abstract] OR “Hearing dog”[Title/Abstract] OR “Hearing dogs”[Title/Abstract] OR “Signal dog”[Title/Abstract] OR “Signal dogs”[Title/Abstract] OR “Medical response dog”[Title/Abstract] OR “Medical response dogs”[Title/Abstract] OR “Seeing eye dog”[Title/Abstract] OR “Seeing eye dogs”[Title/Abstract] )

4-(e.g. 0.035 rather than <0.05) why did researchers choose 0.035 as criterion? In Table 2. Summary of methodological ratings for N=27 studies … etc.

We apologize for this confusion. The method scoring item was “Have actual probability values been reported for most outcomes?” Using e.g. as the abbreviation for the Latin phrase meaning “for example”, we provided 0.035 as an example of exact probability reporting (rather than reporting <0.05). So, 0.035 was not used as any specific criterion, but rather as an example. We revised the text to say “e.g. reporting 0.035 rather than reporting <0.05” to clarify.

The subject of paper is viable, novelty and interested. Systematic review is hard working as I see that in this paper. So, in my opinion paper is acceptable with modification.

We thank the reviewer for their time and consideration reviewing this manuscript. 

Reviewer 2

This article is considering and dealing with an excellent subject which is not only sensitive but also helpful. The authors have thought and looked into the psychological benefits of guide dogs and the importance of a holistic approach on such issues. I am hoping by publishing this article, we would expect to see further research work from the same authors and others.

We thank the reviewer for their positive comments and time reviewing this manuscript.

---

## [Decision Letter · Decision Letter 1]

19 Nov 2020

The effects of assistance dogs on psychosocial health and wellbeing: A systematic literature review

PONE-D-20-22650R1

Dear Dr. Rodriguez,

We’re pleased to inform you that your manuscript has been judged scientifically suitable for publication and will be formally accepted for publication once it meets all outstanding technical requirements.

Kind regards,

Geilson Lima Santana, M.D., Ph.D.

Academic Editor

PLOS ONE

Additional Editor Comments (optional):

Reviewers' comments:

Reviewer's Responses to Questions

**Comments to the Author**

1. If the authors have adequately addressed your comments raised in a previous round of review and you feel that this manuscript is now acceptable for publication, you may indicate that here to bypass the “Comments to the Author” section, enter your conflict of interest statement in the “Confidential to Editor” section, and submit your "Accept" recommendation.

Reviewer #1: All comments have been addressed

2. Is the manuscript technically sound, and do the data support the conclusions?

Reviewer #1: Yes

3. Has the statistical analysis been performed appropriately and rigorously? 

Reviewer #1: Yes

4. Have the authors made all data underlying the findings in their manuscript fully available?

Reviewer #1: Yes

5. Is the manuscript presented in an intelligible fashion and written in standard English?

Reviewer #1: Yes

6. Review Comments to the Author

Reviewer #1: The effects of assistance dogs on psychosocial health and wellbeing: A systematic literature review

Comments

After reading the manuscript several times, I found the authors made instructions given in first review. The objectives of article are clear: line 142 to 150. They detailed them in the context of article.

First aim was starting at line 251

Second aim was starting at line 316

Third objective was starting at line 361.

Article becomes clearly outlined and easy to understand.

Three objectives / aims are discussed in clear way as I suggested in first review.

Search Procedures are rigor and power in the second revision: 156 -172

I scanned the references, I don’t find errors.

I read the whole article, I read reply from authors regarding the first review.

Congratulation for publication

DR. Naser AbdulHafeeth

7. PLOS authors have the option to publish the peer review history of their article (what does this mean?). If published, this will include your full peer review and any attached files.

Reviewer #1: **Yes: **DR. Naser AbdulHafeeth , Taiz University

---

## [Editor Report · Acceptance letter]

20 Nov 2020

PONE-D-20-22650R1 

The effects of assistance dogs on psychosocial health and wellbeing: A systematic literature review 

Dear Dr. Rodriguez:

I'm pleased to inform you that your manuscript has been deemed suitable for publication in PLOS ONE. Congratulations! Your manuscript is now with our production department. 

Kind regards, 

on behalf of

Dr. Geilson Lima Santana 

Academic Editor

PLOS ONE